# Evaluation and Mitigation of Weight-Related Single Event Upsets in a Convolutional Neural Network

Yulong Cai [1], Ming Cai [2], Yanlai Wu [1], Jian Lu [1], Zeyu Bian [1], Bingkai Liu [3,*] and Shuai Cui [1,*]

[1] Innovation Academy for Microsatellites of Chinese Academy of Sciences, Shanghai 200020, China; caiyl@microsate.com (Y.C.)
[2] University of Chinese Academy of Sciences, Beijing 100049, China; caiming@iie.ac.cn
[3] Key Laboratory of Functional Materials and Devices for Special Environments, Xinjiang Technical Institute of Physics and Chemistry, Chinese Academy of Sciences, Urumqi 830011, China
[*] Correspondence: liubk@ms.xjb.ac.cn (B.L.); cuis@microsate.com (S.C.)

**Abstract:** Single Event Upsets (SEUs) are most likely to cause bit flips within the trained parameters of a convolutional neural network (CNN). Therefore, it is crucial to analyze and implement hardening techniques to enhance their reliability under radiation. In this paper, random fault injections into the weights of LeNet-5 were carried out in order to evaluate and propose strategies to improve the reliability of a CNN. According to the results of an SEU fault injection, the accuracy of the CNN can be classified into the following three categories: benign conditions, poor conditions, and critical conditions. Two efficient methods for mitigating weight-related SEUs are proposed, as follows: weight limiting and Triple Modular Redundancy (TMR) for the critical bit of the critical layer. The hardening results show that when the number of SEU faults is small, the weight limiting almost completely eliminates the critical and poor conditions of LeNet-5's accuracy. Additionally, even when the number of SEU faults is large enough, combining the weight limiting and TMR methods for the critical bit of the critical layer can retain the occurrence rate of benign conditions at 98%, saving 99.3% of the hardware resources compared to the Full-TMR hardening method.

**Keywords:** reliability; SEU; CNN; weight



## 1. Introduction

Today, convolutional neural networks (CNNs) are adopted in various domains that range from high-performance computing to data analysis and target recognition for space applications [1–4]. However, when applied in the spaceflight field, CNNs face reliability challenges due to the impact of space radiation. This is because the memory of the hardware accelerators carrying CNNs have been proven to be highly sensitive to energetic particle attacks, especially Single Event Upset (SEU) effects [5–8]. The memory of hardware accelerators stores a large number of weight parameters for CNNs. As process technology scales up, the area consumed by memory dominates the CNN hardware accelerator's area. Consequently, SEUs will become a major reliability concern for future CNN accelerators. Weight-related SEUs will further propagate in CNN computations, ultimately leading to a decrease in CNN accuracy.

Therefore, it is necessary to conduct an assessment of the impact of weight-related SEUs on CNNs and design appropriate hardening measures. Clarifying the fault modes and mechanisms of weight-related SEUs causing CNN accuracy degradation is the basis for studying hardening methods. In [9,10], weight-related SEUs in a CNN were divided into tolerable faults and critical faults, with tolerable faults sometimes referred to as Silent Data Corruption (SDC) [11]. This paper further explores the relationship between weight-related SEUs and the classification accuracy of CNNs, categorizing the impact of weight-related SEUs on CNN accuracy into the following three categories: benign conditions, poor conditions, and critical conditions.

There are many traditional system-level SEU mitigation methods [12,13]; however, due to the large computational requirements and numerous parameters of CNNs, these system mitigation methods are not well suited for CNN hardening needs. On the one hand, the significant number of weight parameters in CNNs occupies a large portion of the storage resources in onboard hardware accelerators, causing the Full-TMR methods to exacerbate the already limited hardware resources. Additionally, error correction coding methods can reduce the CNN's computational speed, while being unable to correct multiple SEU errors.

To overcome the limitations of traditional hardening methods, research on CNN hardening is primarily based on two main approaches. Firstly, it involves improved methods based on the traditional hardening strategies. TMR is a typical method; however, the overhead of TMR can be very high and it can jeopardize the device's performance and efficiency [14,15]. Since not all the errors are critical for a CNN, it is crucial to accurately identify the critical layers of the CNN to balance the utilization of the TMR hardware resources and the hardening effectiveness. In [16], they tested the robustness of the CNN architecture via injecting Gaussian noise into the weights of the CNN and concluded that the error tolerance of a layer tended to worsen as it moved toward the output layer. However, the sensitivity of weight-related SEUs in different layers is influenced by various factors such as the number of weights, the weight value distribution, the network structure, and the layer position. Therefore, this paper characterizes the SEU sensitivity of the weights of different layers in a CNN. Secondly, it involves optimizing the design of CNN algorithms. In [17,18], the method of weight quantization can reduce the sensitivity of CNNs to SEUs. In [19], three modern deep convolutional CNNs were tested for their robustness. It was hypothesized that the use of batch normalization or shortcut connections made ResNet50 and InceptionV3 resilient against memory errors. Some interesting works have proposed adapting the existing Algorithm-Based Fault-Tolerant (ABFT) [20]. However, there is currently no algorithmic optimization hardening method that can significantly improve the SEU resilience of CNN weights.

The main innovations of this paper can be summarized in three points, as follows: firstly, discovering the uncertainty caused by an SEU and classifying the fault modes, with particular emphasis on the possibility of severe consequences, even with a low number of faults; secondly, pointing out that weight-limiting hardening can effectively address this issue, while also improving the rate of benign conditions; and thirdly, proposing a method for identifying critical layers and critical bits and demonstrating the cumulative effects of TMR and weight-limiting hardening. The structure of the remaining sections of this paper is as follows: Section 2 provides an introduction to the LeNet-5 CNN, along with the SEU fault injection method and the simulation results before hardening. In Section 3, we delve into the weight limiting and TMR methods for the critical bit of the critical layer and present their associated outcomes. Finally, Section 4 offers concluding remarks and sets the stage for future work.

## 2. Fault Injection

### 2.1. Simulation Framework

The neural network model used in this work is LeNet-5 and the network structure is shown in Figure 1. We used the Modified National Institute of Standards and Technology (MNIST) dataset, which is composed of hand-written digits, as the input for the LeNet-5 CNN test. The MNIST is commonly used for training various image processing systems or testing in the fields of machine learning.

## LENET-5

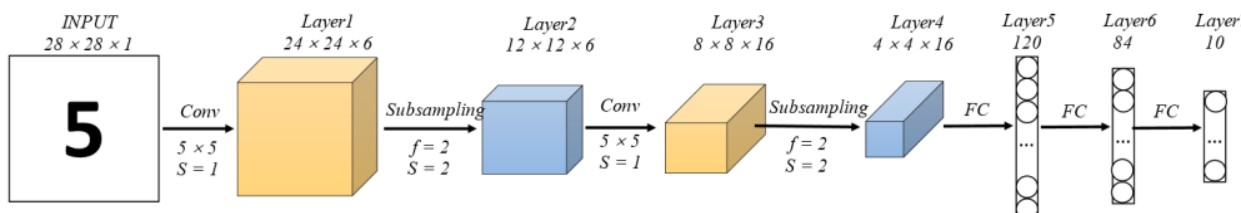

**Figure 1.** The structure of LeNet-5.

Figure 2 represents the framework of simulation. Firstly, the SEUs were introduced to the LeNet-5 after training was complete. Then, the LeNet-5's accuracy was re-evaluated with the new weight. Each weight is represented by 32-bit floating point numbers. The number of bits in SEUs were increased to determine how susceptible the LeNet-5 was to weight-related SEUs and to classify the behavior of the LeNet-5 under an increasing number of SEUs. The bits in SEUs were increased from 0 to 225 and the classification accuracy of LeNet-5 was recorded. In total, 50 trials were recorded for each number of bits in SEUs.

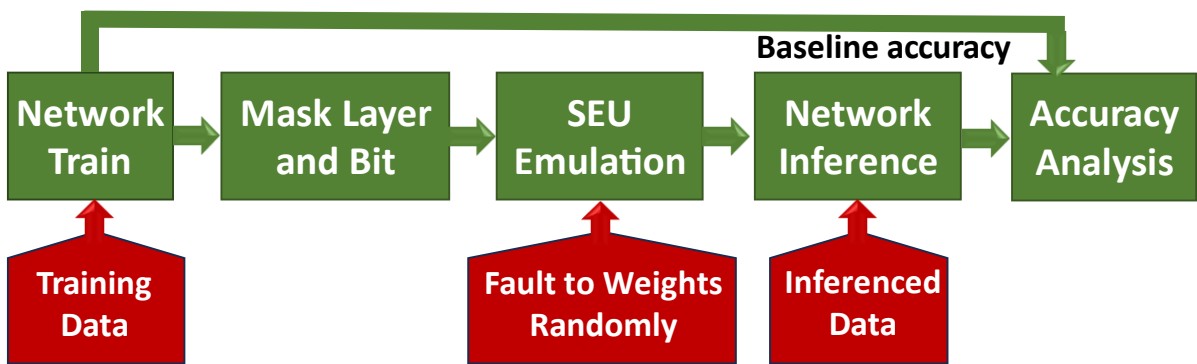

**Figure 2.** Framework for quantifying CNN degradations caused by injecting fault to weights randomly.

The pseudocode for SEU fault injections is shown in the pseudocode Algorithm 1; the theoretical flip rate for each bit is first calculated based on the bit flip rate of the weight storage unit and the duration of the on-orbit operation. Then, all weight bits are traversed, each being flipped with a probability of p. The number of SEU faults injected is a statistic obtained by multiplying the SEU rate by the total number of weights. To simulate the real-world scenario where the same bit can be flipped multiple times, the on-orbit duration is divided into batch-size parts, with faults being injected for a duration of years/batch size each time. Additionally, the simulation models the situation where multiple bit flips are caused by the accumulation of SEUs; however, it does not simulate the scenario where multiple bit flips are caused by a single particle radiation. Furthermore, TMR (Triple Modular Redundancy) and weight limit hardening have been set as adjustable options, making it convenient to obtain the effects of different hardening combinations. The purpose of this is to provide more flexibility, allowing the most suitable hardening strategy to be chosen based on different requirements and conditions.

---

**Algorithm 1** Inject-Fault

---

**Input:** mod-el, datatype, TMR, p, years, limitvalid, batchsize
**Output:** model
/* model: LENET-5 model.                                                              */
/* datatype: Data type for model's weights and bias.                                  */
/* TMR: Zero-TMR, Partial-TMR and Full-TMR.                                            */
/* p: The probability of flipping a single bit per year.                              */
/* years: The duration of exposure for devices in the radiation space.                */
/* limitvalid: Flag for limiting the weights after fault injection.                   */
/* batchsize: Inject faults in multiple batches.                                      */

1.   *weights=model.GetWeights()*

2.   **if** *datatype = = "float32"* **then**

3.      **for** $t \leftarrow 0$ to $\frac{years}{batchsize}$ **do**

4.         *p = batchsize * p*

5.         **for** *index $\leftarrow$ 0 to len(weights)* **do**

6.            *data = float32_to_bin(weight[index])*

7.            **for** *dindex $\leftarrow$ 0 to len(data)* **do**

8.               **if** *NoTMR(TMR, dindex)* **then**
                      //the function SeuRandom is used to generate an enable signal
                      with a probability of p being effective

9.                  *Flip(data[dindex], SeuRandom(p))*

10.              **else**

11.                 *flip(data[dindex], SeuRandom(PTMR(p)))*

12.           *Weights[index] = bin_to_float32(data)*

13.           // the function isnan is used to check if a number is invalid

14.           **if** *isnan(weights[index])* then

15.              *weights[index] = 0*

16.           **else**

17.              **if** *limitvalid and (weights[index]>uplimit)* then

18.                 *weights[index] = weights.max() >> 1*

19.              **else if** *limitvalid and (weights[index]<lowlimit)* then

                     *Weights[index] = weights.min() >> 1*

20.  **else**
           // operations of other datatype similar to float32

21.      ...

22.  *Model.SetWeights(reshape(weights))*

---

### 2.2. Results before Hardening

Before hardening, the accuracy of LeNet-5 varies with the number of SEU fault injections, as shown in Figure 3. Each column of 50 blue dots represents 50 trials, with brighter colors indicating a higher frequency of occurrences. From the graph, the following

three phenomena can be observed: First, despite the same number of SEU fault injections, there is a significant difference in LeNet-5's accuracy. Second, there is a high frequency of extremely low LeNet-5 accuracy (below 0.3), even when the number of SEU fault injections is small, with a 31% probability of critical situations occurring. Third, as the number of SEU fault injections increases, the maximum accuracy of LeNet-5 rapidly decreases and, when the number of SEU fault injections reaches 125, almost all classification accuracies are below 0.85. Next, we will delve into the reasons behind these three phenomena, propose corresponding efficient hardening methods, and validate the effectiveness of the hardening measures through SEU fault simulation.

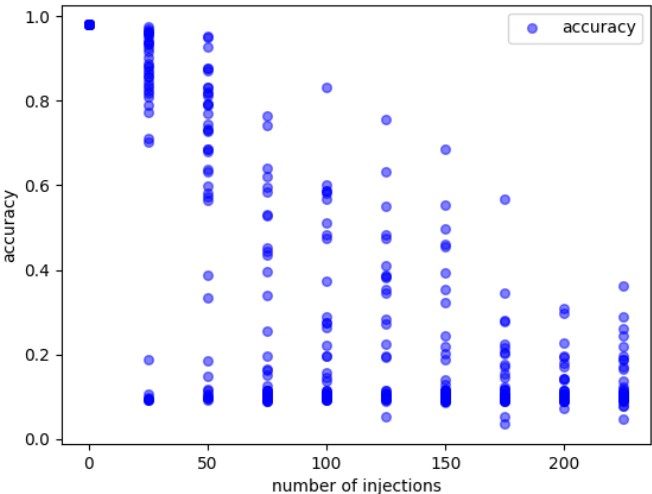

**Figure 3.** LeNet-5's accuracy varies with the number of SEUs injected.

## 3. Hardening Methods and Validation

Based on the changes in CNN accuracy and the acceptance criteria for specific spaceflight tasks, we categorize the SEU-induced CNN accuracy into the following three categories: benign, poor, and critical. The benign condition refers to an accuracy higher than the acceptable accuracy for spaceflight tasks. In this paper, we define an accuracy greater than 85% as a benign condition. The poor condition refers to an accuracy between the accuracy of blind guessing and the acceptable accuracy for spaceflight tasks. In this paper, we define an accuracy between 30% and 85% as a poor condition. The critical condition refers to an accuracy lower than the accuracy of blind guessing. Figure 4 illustrates the relationship between the three types of LeNet-5 accuracy and the number of SEU fault injections in the absence of hardening. With an increase in the number of SEU faults, the benign condition gradually decreases. When the number of SEU faults reaches around 50, the poor condition has the highest proportion. When the number of faults increases to 75, the critical condition already accounts for 70% of cases, indicating that LeNet-5 is almost unable to function properly. To improve CNN accuracy, the primary goal is to minimize the occurrence of critical conditions. It is important to note that the CNN accuracy classification method provided in this manuscript is universal, but the threshold conditions and spaceflight task requirements for different classifications (benign, poor, and critical) are closely related. Different task requirements will affect the threshold values. In our engineering code, the threshold for determining "benign" and "poor" conditions is designed as a variable.

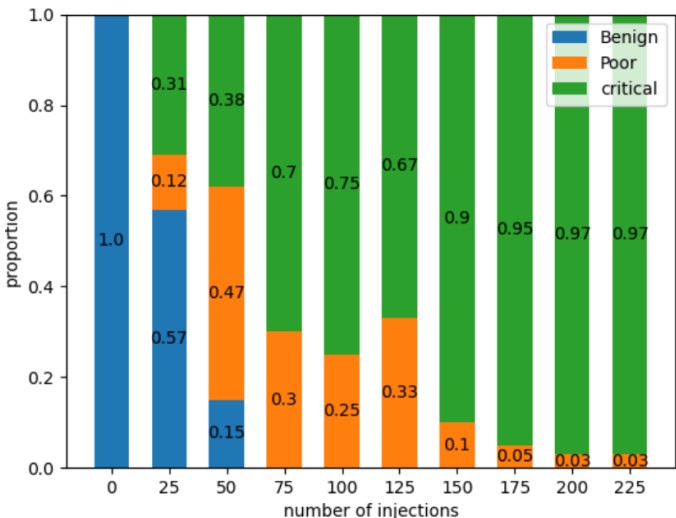

**Figure 4.** The proportion of the three LeNet-5 accuracy conditions caused by weight-related SEUs.

The statistical fault injection method in a CNN was discussed [21], and the fault injection results of this project were statistically analyzed under experimental conditions. The abscissa in Figure 4 represents the number of fault injections. We will separately analyze the distribution of the network model for different numbers of fault injections to reflect the dynamic changes of the network in orbit. After specifying the number of fault injections x, the sample population of fault injections becomes all combinations of selecting x bits from all weight bits. In our experiment, the sample size was selected as 50. For each combination, when the selected weights are flipped, the CNN inferences a test set of 10 k images and obtains a final accuracy. Then, based on the accuracy, the CNN output situation is classified (for each specific classification, it is a binomial distribution). Therefore, in the case of sampling sample size n, the standard error of the proportion of one output situation of CNN is:

$$SE(p) = \sqrt{\frac{p \times (1-p)}{n}} \tag{1}$$

where p represents the proportion of one output situation of the CNN in the sampled sample. The confidence interval is:

$$p \pm z \times SE(p) \tag{2}$$

where z and confidence level are related. According to Figure 4, when the injection count is 25 (benign situation accounts for 0.57), the sample size is 50, the confidence level is 90%, the standard error of the benign situation is 0.07, and the confidence interval is [0.45, 0.68]. The errors and confidence interval conditions of the results at other injection counts are even better than this. We believe that such results are acceptable.

### 3.1. Weight Limiting

Research has shown that the distribution of weights in CNNs follows an approximately normal distribution with a mean of 0, indicating that the majority of weight values are small [22]. The weight distributions of the LeNet-5 network are shown in Figures 5 and 6, where Figure 5 represents the weight distribution of the second convolutional layer and Figure 6 represents the weight distribution of the first fully connected layer. Similar patterns can be observed in the weight distributions of other convolutional and fully connected layers.

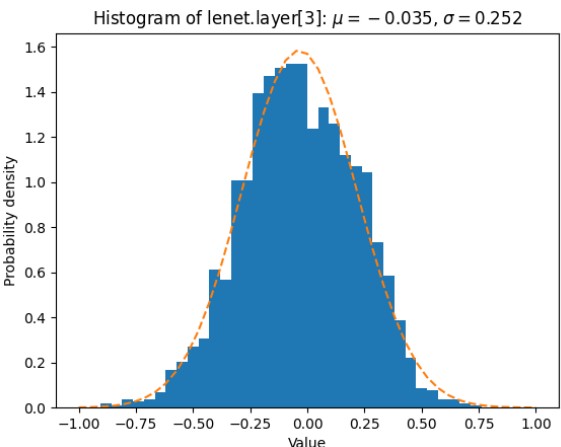

**Figure 5.** The weight distributions of the third layer of the LeNet-5 network.

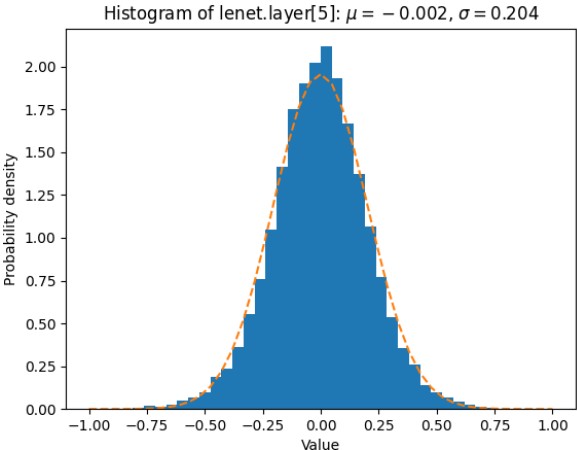

**Figure 6.** The weight distributions of the sixth layer of the LeNet-5 network.

Therefore, weight-related SEUs can potentially cause extreme weight values and, in this study, weights that exceed twice the normal weight value are defined as extreme weights. Extreme weights can have a significant impact on the layer in which they are present; when extreme weights occur in the convolutional kernels of a convolutional layer, the other weights in that kernel no longer function, leading to the failure of an output channel in that convolutional layer. Similarly, when extreme weight values occur in a specific row of the weight matrix in a fully connected layer, the corresponding output neuron in that row becomes ineffective. Moreover, errors caused by the changing weights propagate backward through the network, so even a small number of extreme weights can cause the entire network to degrade into a critical condition.

Next, we will establish a model to quantitatively evaluate the probability of extreme weights, leading to the degradation of CNN accuracy into a critical condition after T years. This model primarily considers the following two factors: the probability of SEUs causing weights to become extreme and the probability of extreme weights leading to a critical condition.

The probability p1 is defined as the actual SEU probability of a bit storing weights in the hardware accelerator. Considering that a bit can be affected by multiple SEUs, we can calculate the actual SEU probability p2 of a bit after T years using Formula (3), and the probability p3 represents that a weight has flip bit ($\geq$1 bit), where bitlen represents the bit width of the data type:

$$p2 = \sum_{2 \nmid d, d \leq T} C_T^d p1^d (1 - p1)^{T-d} \tag{3}$$

$$p3 = 1 - C_{bitlen}^0 \times (1 - p2)^{bitlen} \tag{4}$$

Then, we analyze the probability of SEUs causing a weight to become extreme. The weights discussed in this study are represented as single-precision floating-point numbers, which consist of three parts, as follows: sign bit, exponent, and mantissa. Once a high-order bit of the exponent is affected by an SEU, it can have a significant impact on the value of the floating-point number (IEEE 754), as shown in the example in Figure 7. To obtain the probability of a weight becoming extreme, we first generate a set of data that follows a normal distribution, based on the mean and standard deviation of the weights. Then, we calculate the probability that each number in this set becomes an extreme weight due to SEUs and take the arithmetic average of these probabilities. The resulting probability is 6.7%, which we denote as p4.

**Figure 7.** Example of an SEU causing an extreme weight.

p5[x] represents the probability that the number of extreme weights is x after T years. p6[x] represents the probability of the model becoming a critical condition when there are x extreme weights in the CNN.

$$p5[x] = C_N^x (p3 * p4)^x (1 - p3 * p4)^{N-x} \tag{5}$$

$$p6[x] = \frac{\sum_{i=0}^{n-1} \text{flag\_iter}[i]}{n}, \text{flag\_iter}[i] \in \{0, 1\} \tag{6}$$

In the equation, flag_iter[i] represents whether the CNN test result of the i-th iteration is of a critical condition and n represents the number of random injection tests performed for each number of extreme weights. p represents the probability that the CNN reaches a critical condition in the Tth year:

$$p = \sum_{x=0}^{N} p5[x]p6[x] \tag{7}$$

Through Formula (7), we found that when no hardening is performed, the probability of extreme weights causing LeNet-5's accuracy to reach a critical condition is as high as 99% when the number of SEU fault injections reaches 225. Therefore, we can conclude that 6.7% of SEUs dominate the 97% critical condition in Figure 2, while the remaining 93.3% of SEUs account for the 0.03% poor condition. Additionally, when the number of SEUs is below a certain value, both poor conditions and critical conditions are mainly caused by extreme weights.

Based on the characteristics of weight distribution and the main causes of LeNet-5's accuracy degradation, we propose a new hardening method, weight limiting. The principle is shown in the following equation:

$$w_{buffer} = \begin{cases} 0 & w\_men \in \{nan, inf\} \\ wmax \gg 1 & w\_mem \leq uplimit \\ wmin \gg 1 & w\_mem \geq lowlimit \end{cases} \tag{8}$$

In the equation, w_buffer represents the weights in the buffer closest to the calculation unit, w_mem represents the weights stored at the previous level of w_buffer, nan (not a number) and inf (infinite) are used to determine if the current weights are invalid, and uplimit and lowlimit are the upper and lower limits, respectively. The variables wmax and wmin represent the maximum and minimum weight values, respectively. The symbol ">>" represents a right shift of 1 bit. This means that the binary representation of a number is shifted to the right by one position, effectively dividing the number by 2.

In Figures 8 and 9, weight limiting and Full-TMR are compared as two hardening methods. Figure 8 compares the probability of LeNet-5's accuracy reaching a critical condition, while Figure 9 compares the probability of LeNet-5's accuracy reaching a benign condition. The results in Figure 8 indicate that the weight limiting method can effectively eliminate critical conditions, as extreme weights are the main cause of critical conditions. From Figure 9, it can be observed that when the number of SEUs is less than 380, weight limiting significantly improves the proportion of LeNet-5's benign accuracy, reaching as high as 98%. So extreme weights are also the dominant cause of poor conditions at this stage. As the number of SEUs further increases, the probability of critical conditions remains close to 0% under the weight limiting, but the benign condition starts to degrade. This is because the limiting measures cannot eliminate the small range fluctuations in weights caused by SEUs. Moreover, with an increasing number of SEU faults, the accumulation of small range fluctuations in weights gradually has a greater impact on LeNet-5's accuracy.

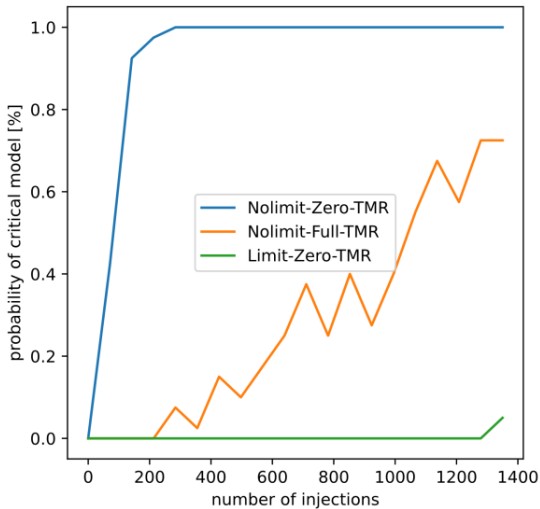

**Figure 8.** The probabilities of critical condition change with the number of SEUs under different hardening methods.

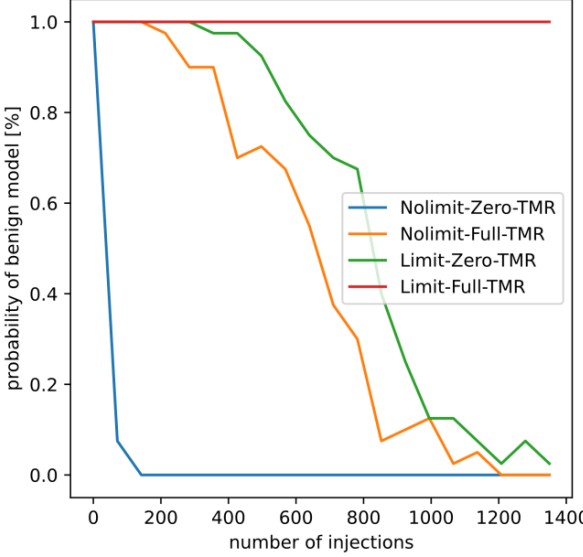

**Figure 9.** The probabilities of benign condition change with the number of SEUs under different hardening methods.

Figure 9 shows that when weight limiting is combined with Full-TMR, it can effectively address the issue of declining benign conditions. However, the Full-TMR method consumes

a large amount of hardware resources and is not suitable for scenarios with a large number of weights, such as CNNs. Therefore, it becomes particularly important to select key positions for Partial-TMR.

### 3.2. Selective TMR

The TMR process for the critical bits of CNN weights can be divided into two steps. The first step is to identify the critical weight layers. The second step is to harden the critical layers by protecting the sign bit, the most significant bit of the exponent, and the most significant bit of the mantissa of the weights in those layers. It is particularly important to note that the weight limiting method has already been applied to LeNet-5 in the following simulation.

To identify the critical layers of LeNet-5, first, TMR is applied to the weights of different layers individually. Then, SEU faults are randomly injected into the weights of all layers; the accuracy results of LeNet-5 are shown in Figure 10. Figure 10 shows the results of TMR hardening applied to the weights of each layer in LeNet-5 and there is no apparent dependency on the effectiveness of TMR across different layers. Considering the number of weights in each layer, as given in Table 1, the first layer, the third layer, and the seventh layer show the highest efficiency in hardening. This means that significant improvement in LeNet-5's accuracy can be achieved by applying TMR to a small number of weights in these three layers. Therefore, these three layers are defined as the critical layers. Figure 11 compares the hardening results of all layers, critical layers, convolutional layers, and fully connected layers. The results show that the TMR of the critical layers has a significantly better effect than the convolutional layers and fully connected layers.

Next, we compare the following five scenarios: TMR for full-bit of all layers (all-all-TMR), TMR for full-bit of the critical layer(partial-all-TMR), TMR for the critical bit of all layers (all-partial-TMR), TMR for the critical bit of the critical layer (partial-partial-TMR), and no TMR (Zero-TMR). The simulation results are shown in Figure 12. It is evident that the best effect is achieved with TMR for the full-bit of all layers, followed by the full-bit of the critical layer (92.3% reduction in resources) and the critical bit of all layers (90.6% reduction in resources). Next is the critical bit of the critical layer (99.3% reduction in resources) and, finally, no TMR (100% reduction in resources).

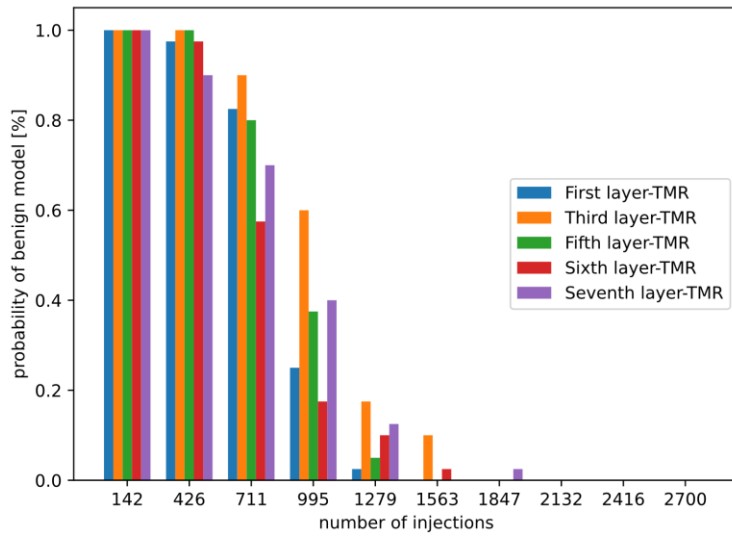

**Figure 10.** The probability of benign conditions after hardening for individual layers in LeNet-5.

**Table 1.** The exact number of weights in each layer of LeNet-5.

| Layer | All | Layer 1 | Layer 2 | Layer 3 | Layer 4 | Layer 5 | Layer 6 | Layer 7 |
|---|---|---|---|---|---|---|---|---|
| Number of weights | 44,426 | 156 | 0 | 2416 | 0 | 30,840 | 10,164 | 850 |

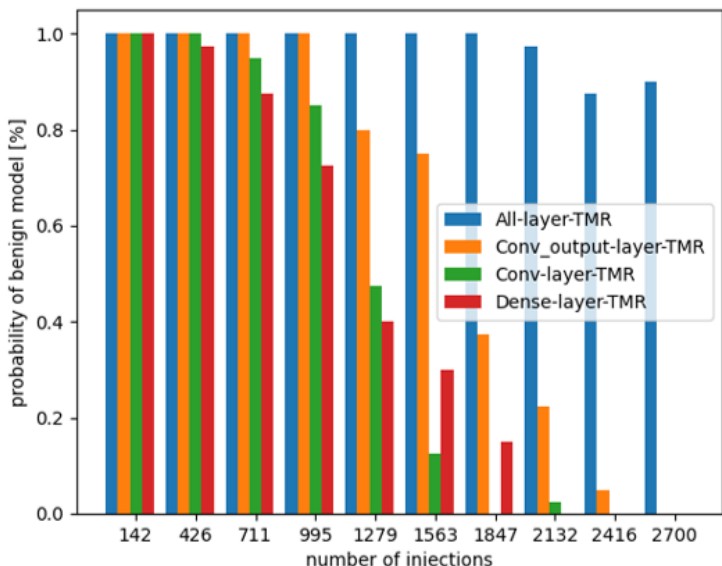

**Figure 11.** The probability of benign conditions after hardening for the combination of multiple layers in LeNet-5.

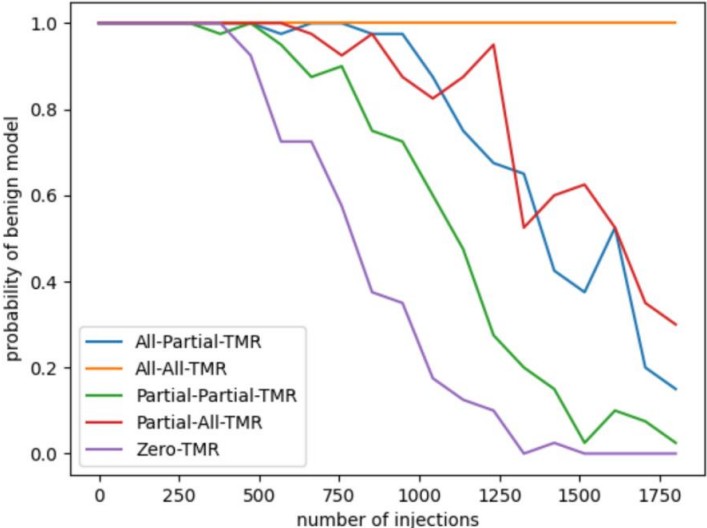

**Figure 12.** Comparison of the effectiveness of five different selective TMR methods.

The choice of the final hardening method depends on factors such as the mission environment, radiation environment, on-orbit duration, device sensitivity, CNN accuracy classification criteria, network model, dataset, hardware resources, and task requirements. Under the given conditions in this paper, when the number of on-orbit SEU fault injections is less than 600, selecting the partial-partial-TMR hardening method (weight limiting method is already applied by default) can maintain the benign condition probability of LeNet-5 at 98%.

## 4. Conclusions

In this paper, it was discovered that impact of weight-related SEUs on CNN accuracy has significant uncertainty and, even at very low SEU fault injection rates, extremely low accuracy can occur. This phenomenon indicates that the CNN accuracy obtained from a single irradiation experiment cannot represent its true accuracy. Based on CNN accuracy and specific space mission requirements, we classify the CNN accuracy into the following three categories: benign conditions, poor conditions, and critical conditions.

This paper analyzes the reasons for the degradation in CNN accuracy from two aspects. First, SEUs lead to extreme weights, which significantly affect the layer where the weights are located. Second, SEUs cause small variations in weight values and, only when a certain number of such weights accumulate, can they lead to poor or critical CNN conditions. We establish a mathematical model to quantitatively evaluate the probability of extreme weights causing CNN accuracy to degrade into a critical condition. The model shows that the probability of SEUs causing weights to become extreme is 6.7%. When the number of SEU fault injections reaches 225, the probability of extreme weights leading to CNN accuracy reaching critical condition reaches 99%.

Two hardening methods are proposed, as follows: weight limiting and partial-partial-TMR. When the number of SEU faults is below a certain threshold, weight limiting reduces the probabilities of critical and poor conditions in LeNet-5 to almost zero, because extreme weights dominate the accuracy degradation. As the number of SEU faults increases, the accumulation of small variations in weights starts to affect the benign condition. We identify the critical layers of LeNet-5 as the input for Selective TMR, which are the first layer, third layer, and seventh layer. Under the given application conditions in this paper, when the number of SEU faults is large, both weight limiting and partial-partial-TMR methods maintain a benign condition occurrence rate of approximately 98%, while reducing TMR resource consumption by 99.3%. It is important to note that the quantitative analysis conclusions in this paper may vary depending on the CNN type, accuracy classification criteria, and definition of extreme weights. However, the qualitative analysis conclusions apply to all CNN weight flips.

**Author Contributions:** Conceptualization, Y.C.; methodology, S.C. and B.L.; software, M.C.; validation, Y.W.; formal analysis, Z.B.; writing—original draft preparation, J.L.; writing—review and editing, Y.C. All authors have read and agreed to the published version of the manuscript.

**Funding:** This research was funded by the National Natural Science Foundation of China, grant number 12305323" and, in part, by the Innovation Foundation of Radiation Application, China Institute of Atomic Energy, under Grant KFZC2022020101.

**Data Availability Statement:** The original data presented in the study are openly available in GitHub at https://github.com/minghan1/Letnet-seu.git, accessed on 1 March 2023.

**Conflicts of Interest:** The authors declare no conflicts of interest.

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
