# Peer review of "Evaluation and Mitigation of Weight-Related Single Event Upsets in a Convolutional Neural Network"

_electronics, doi:10.3390/electronics13071296_

Round 1

Reviewer 1 Report

Comments and Suggestions for Authors

Dear Authors, 

I have read with interest your paper. The topic addressed is really interesting and relevant for the technological advancement of space and space-like electronic systems. 

I have a few comments on the paper:

1) Row 80: I imagnine that you are referring to the "Modified" National Institute of Standard and Technology dataset (MNIST) and not "Mixed"

2) Rows 110-111: (Figure 3) The algorithm is almost unreadable in my A4 600DPI printout. I suggest you to increase the character size, if possible.

3) Rows 131-132: There is a reference for the definition of a "benign" condition and accuracy greater than 85% ? This value comes out from a standard or a best practice ?

4) Rows 171 - 200: I understand the general principle of your model but I found some problems with the explanation. In particular you define the probability p1 in the text (row 172) but p1 is not present in any of the following two equations.   I think you should review this paragraph verifying expression and check if all symbols used are described in the text.  Also the caption of Figure 8 is wrong. 

5) Rows 208 - 215: The text is somewhat confusing for me. What do you mean by wmax>>1 and wmin<<1 ? I suggest you to rewrite the paragraph describing in more detail the method.

Regards 

Author Response

1)

Mixed" has been corrected to "Modified.

2)

The image format has been changed to vector graphic.

3)

It is important to note that the output accuracy classification method provided in this manuscript is universal, but the threshold conditions and spaceflight task requirements for different classifications (benign, poor, critical) are closely related. Different task requirements will affect the threshold values. (The above sentence is added to rows 143 of the manuscript to further explain the principles of threshold selection.)

4)

After checking the symbols of all the formulas, it was found that there were errors in the symbols in formulas (1), (2), and (3), and these have all been corrected. The caption of Figure 8 has been corrected to "Example of SEU causing extreme weight.

5)

In equation (6), the symbol “<<” is corrected to “>>”. the symbol ">>" represents a right shift of 1 bit. This means that the binary representation of a number is shifted to the right by one position, effectively dividing the number by 2. The variables wmax and wmin represent the maximum and minimum weight values respectively. ( The above sentence is added to rows 136 of the manuscript)

Reviewer 2 Report

Comments and Suggestions for Authors

Major comments

Benign condition, poor condition, and critical condition should be much better defined in this paper, not just giving a reference. This is crucial for correct understanding of the author's message. Also, what is the justification for the 85% limit as a benign condition? Why not 90% or 80%?

Weight distributions of the layers: this should depend on the learning phase. What was the learning conditions for the presented weights? More globally, the paper lacks precise information about what is shown or discussed.

Most comments correspond to quite obvious well-known facts. As an example, assuming a large number of SEUs (much higher than what is usually considered inacceptable) the accuracy of LeNet falls down - this is not a breakthrough ... even if the conclusion says "it was discovered that ...". It is the same for Fig. 5.

section 3.1: the discussion on probabilities is absolutely not clear. Once again the limits about the level of criticality are not justified. In addition, how is the relationship between e.g., "critical weights" and "critical conditions" evaluated?

What does "the buffer closest to the calculation unit" mean? In this paper there is no notion of target microarchitecture or even architecture so no specific buffer can be considered and furthermore thera is no possibility toidentify any distance to any calculation unit.

The conditions (experimental procedure) for fault injections should be cleraly stated. There is no information about the number of injections, the error margin and confidence, the uniformity of the random selection, etc. Also, this procedure is important to be sure that the results are exploitable. See for example:
A. Ruospo∗, G. Gavarini∗, C. De Sio∗, J. Guerrero∗, L. Sterpone∗, M. Sonza Reorda∗, E. Sanchez∗, R. Mariani†, J. Aribido† and J. Athavale†
Assessing Convolutional Neural Networks Reliability through Statistical Fault Injections
2023 Design, Automation & Test in Europe Conference (DATE 2023)

Due to all lacking information the main conclusion about the hardening effects cannot be convincing.

I think this paper should be noticeably re-worked and completed.

Minor comments

Fig. 4: the text mentions "with darker colors indicating a higher frequency of occurrences". This is not visible.

I don't understand the use of [12] as reference about CNNs

English should be noticeably revised. Proofreading is also required (e.g., "2.2. Results before harding").

Comments on the Quality of English Language

English should be noticeably revised.

Author Response

Benign condition, poor condition, and critical condition should be much better defined in this paper, not just giving a reference. This is crucial for correct understanding of the author's message. Also, what is the justification for the 85% limit as a benign condition? Why not 90% or 80%?

Answer: It is important to note that the output accuracy classification method provided in this article is universal, but the threshold conditions and engineering task requirements for different classifications (benign, poor, critical) are closely related. Different task requirements will affect the threshold values. In our engineering code, the threshold for determining "benign" and "poor" conditions is designed as a variable. (The above sentence is added to rows 143 of the manuscript to further explain the principles of threshold selection.)

Weight distributions of the layers: this should depend on the learning phase. What was the learning conditions for the presented weights? More globally, the paper lacks precise information about what is shown or discussed.

Answer: The distribution of weights is related to the learning phase, but according to reference [21], the weights of a well-trained convolutional neural network have a Gaussian-like distribution with a mean of 0. The first and second paragraphs of Section 3.1 of the manuscript provide detailed discussions on this.

[21] Z. Huang, W. Shao, X. Wang, L. Lin, and P. Luo, “Rethinking the pruning criteria for convolutional neural network,” in Neural Information Processing Systems, 2021.
Most comments correspond to quite obvious well-known facts. As an example, assuming a large number of SEUs (much higher than what is usually considered inacceptable) the accuracy of LeNet falls down - this is not a breakthrough ... even if the conclusion says "it was discovered that ...". It is the same for Fig. 5.

Answer: Regarding the example you mentioned, our expression has some issues. What we want to convey is that the impact of weight SEU on CNN accuracy has significant uncertainty (The above sentence is added to rows 313 of the manuscript). The main innovations of this manuscript can be summarized in three points:

  1. Discovering the uncertainty caused by SEU and classifying fault modes, with particular emphasis on the possibility of severe consequences even with a low number of faults.
  2. Pointing out that weight limiting hardening can effectively address this issue while also improving the rate of benign conditions.
  3. Proposing a method for identifying critical layers and critical bits, and demonstrating through experiments the cumulative effects of TMR hardening and weight limiting hardening.

section 3.1: the discussion on probabilities is absolutely not clear. Once again the limits about the level of criticality are not justified. In addition, how is the relationship between e.g., "critical weights" and "critical conditions" evaluated?

Answer: After checking the symbols of all the formulas, it was found that there were errors in the symbols in formulas (1), (2), and (3), and these have all been corrected. The error in the symbol leads to that the discussion on probabilities is absolutely not clear.

Like equation (6), in each experiment, we randomly select x weights to become extreme weights, and then calculate the proportion of extreme cases in these n trials of the CNN.

What does "the buffer closest to the calculation unit" mean? In this paper there is no notion of target microarchitecture or even architecture so no specific buffer can be considered and furthermore thera is no possibility toidentify any distance to any calculation unit.

Answer: The article does not discuss specific hardware circuits, but considers the practical application of weight limiting hardening methods. When applying this weight limiting method in actual circuits, we suggest that designers implement weight limiting hardening in the weight cache closest to the computing unit to reduce the hardware resources used.

The conditions (experimental procedure) for fault injections should be cleraly stated. There is no information about the number of injections, the error margin and confidence, the uniformity of the random selection, etc. Also, this procedure is important to be sure that the results are exploitable. See for example:
A. Ruospo∗, G. Gavarini∗, C. De Sio∗, J. Guerrero∗, L. Sterpone∗, M. Sonza Reorda∗, E. Sanchez∗, R. Mariani†, J. Aribido† and J. Athavale†
Assessing Convolutional Neural Networks Reliability through Statistical Fault Injections
2023 Design, Automation & Test in Europe Conference (DATE 2023)

Answer: The statistical fault injection method in CNN was discussed, and the fault injection results of this project were statistically analyzed under the experimental conditions. The abscissa in Figure 5 represents the number of fault injections. We will separately analyze the distribution of the network model for different numbers of fault injections to reflect the dynamic changes of the network in orbit. After specifying the number of fault injections x, the sample population of fault injections becomes all combinations of selecting x bits from all weight bits. In our experiment, the sample size is selected as 50. For each combination, when the selected weights are flipped, the CNN inference contains a test set of 10k images and obtains a final accuracy. Then, based on the accuracy, the CNN output situation is classified (for each specific classification, it is a binomial distribution). Therefore, in the case of sampling sample size n, the standard error of the proportion of one output situation of CNN is:

where p represents the proportion of one output situation of CNN in the sampled sample. The confidence interval is:

The selection of z and confidence level in the equation is related. According to Figure 5, when the injection count is 25 (benign situation accounts for 0.57), the sample size is 50, and the confidence level is 90%, the standard error of the benign situation is 0.07, and the confidence interval is [0.45, 0.68]. The errors and confidence interval conditions of the results at other injection counts are even better than this. We believe that such results are acceptable.

(The above sentence is added to rows 148-168 of the manuscript)

Due to all lacking information the main conclusion about the hardening effects cannot be convincing.

Answer: The engineering files of the simulation are provided as attachments, and we further enhance the specific conditions and procedures of the simulation.

I think this paper should be noticeably re-worked and completed.

Answer: The issues raised by the reviewer have been carefully considered and addressed.

Minor comments

Fig. 4: the text mentions "with darker colors indicating a higher frequency of occurrences". This is not visible.

Answer: "with darker colors indicating a higher frequency of occurrences "has been corrected to" with brighter colors indicating a higher frequency of occurrences "

I don't understand the use of [12] as reference about CNNs

Answer: Reference [12] is used to introduce traditional hardening methods.
English should be noticeably revised. Proofreading is also required (e.g., "2.2. Results before harding").

English should be noticeably revised. Proofreading is also required (e.g., "2.2. Results before harding"). 

Answer: The language in the manuscript has been polished again.

"2.2. Results before harding" has been corrected to" hardening "

Reviewer 3 Report

Comments and Suggestions for Authors

The paper presents theoretical analysis of how CNNs are impacted by SEUs and explores how hardening strategies such as weight limiting and TMR can further help improve the radiation tolerance. Random fault injections are carried out in the weights of LeNet 5 and depending on the author defined CNN accuracy the efficacy of the proposed hardening techniques are evaluated. Generally speaking the paper is well written and the conclusions drawn support the experimental analysis. However, a few typos need to be corrected, such as, MNIST is Modified (not Mixed) National Institute of Standards and Technology Database. Recommend a careful reading to correct these. Other than this, the reviewer has no significant concerns.

Author Response

Carefully checked the manuscript for typos, including the following issues:

  1. Mixed" has been corrected to "Modified.
  2. "2.2. Results before harding" has been corrected to" hardening".
  3. The caption of Figure 8 has been corrected to "Example of SEU causing extreme weight.

Round 2

Reviewer 2 Report

Comments and Suggestions for Authors

The message given in the paper has been improved in this new version. Some aspects may still be improved, including the exact application of Formula 8, but the global idea can be better catched.

I think that the answer about "the buffer closest to the calculation unit" should have been reported in some way in the paper. Also, the following answer may have been better emphasized in the introduction:
"The main innovations of this manuscript can be summarized in three points:
    Discovering the uncertainty caused by SEU and classifying fault modes, with particular emphasis on the possibility of severe consequences even with a low number of faults.
    Pointing out that weight limiting hardening can effectively address this issue while also improving the rate of benign conditions.
    Proposing a method for identifying critical layers and critical bits, and demonstrating through experiments the cumulative effects of TMR hardening and weight limiting hardening."

I maintain that ref. [12] (and even [13]) is not a right reference where it is cited ("traditional system-level memory SEU mitigation methods are not well suited for CNN"). Or it should be cited after "traditional system-level memory SEU mitigation methods" before mentioning CNNs.

Author Response

  1. This manuscript provides a method for weight limit control and simulation results. The description of the relevant method in the manuscript can support engineers in carrying out practical applications.
  2. The below sentence is added to rows 73-79 of the manuscript: The main innovations of this manuscript can be summarized in three points: Firstly, discovering the uncertainty caused by SEU and classifying fault modes, with particular emphasis on the possibility of severe consequences even with a low number of faults; Secondly, pointing out that weight limiting hardening can effectively address this issue while also improving the rate of benign conditions; Thirdly, Proposing a method for identifying critical layers and critical bits, and demonstrating the cumulative effects of TMR hardening and weight limiting hardening.
  3. ref. [12] and [13] have been cited after "traditional system-level memory SEU mitigation methods" before mentioning CNNs.